# Ca^2+^–Calmodulin Dependent Wound Repair in *Dictyostelium* Cell Membrane

**DOI:** 10.3390/cells9041058

**Published:** 2020-04-23

**Authors:** Md. Shahabe Uddin Talukder, Mst. Shaela Pervin, Md. Istiaq Obaidi Tanvir, Koushiro Fujimoto, Masahito Tanaka, Go Itoh, Shigehiko Yumura

**Affiliations:** 1Graduate School of Sciences and Technology for Innovation, Yamaguchi University, Yamaguchi 753-8511, Japan; shahabeuddin@yahoo.com (M.S.U.T.); sprvn@yahoo.com (M.S.P.); tanviristiaq@gmail.com (M.I.O.T.); i501wz@yamaguchi-u.ac.jp (K.F.); g003wf@yamaguchi-u.ac.jp (M.T.); 2Institute of Food and Radiation Biology, AERE, Bangladesh Atomic Energy Commission, Savar, Dhaka 3787, Bangladesh; 3Rajshahi Diabetic Association General Hospital, Luxmipur, Jhautala, Rajshahi 6000, Bangladesh; 4Department of Molecular Medicine and Biochemistry, Akita University Graduate School of Medicine, Akita 010-8543, Japan

**Keywords:** actin, calcium ion, calmodulin, cell membrane, *Dictyostelium*, wound repair

## Abstract

Wound repair of cell membrane is a vital physiological phenomenon. We examined wound repair in *Dictyostelium* cells by using a laserporation, which we recently invented. We examined the influx of fluorescent dyes from the external medium and monitored the cytosolic Ca^2+^ after wounding. The influx of Ca^2+^ through the wound pore was essential for wound repair. Annexin and ESCRT components accumulated at the wound site upon wounding as previously described in animal cells, but these were not essential for wound repair in *Dictyostelium* cells. We discovered that calmodulin accumulated at the wound site upon wounding, which was essential for wound repair. The membrane accumulated at the wound site to plug the wound pore by two-steps, depending on Ca^2+^ influx and calmodulin. From several lines of evidence, the membrane plug was derived from de novo generated vesicles at the wound site. Actin filaments also accumulated at the wound site, depending on Ca^2+^ influx and calmodulin. Actin accumulation was essential for wound repair, but microtubules were not essential. A molecular mechanism of wound repair will be discussed.

## 1. Introduction

The survival of single cells is largely dependent on their cell membrane. The cell membrane has ability to protect the cell from extracellular harmful effectors such as physical damage, adverse environmental conditions, and pathogens. The cell membrane is also damaged during normal physiological functions such as muscle contraction [1,2]. Studies involving cell membrane repair mechanisms are getting attention as it is related to various diseases including diabetes [3], muscular dystrophies [4,5], acute kidney injury [6], and vitamin deficiencies [7]. In addition, many methods for introducing extracellular substances into cells, including microinjection and electroporation, rely on the cellular ability of wound repair.

Researchers have developed various models and mechanisms related to wound repair in several model organisms such as *Drosophila* [8,9,10,11], Yeast [12], *C*. *elegans* [13], *Xenopus* [14], cultured animal cells [15,16,17,18,19], and *Dictyostelium* cells [20]. A common feature among them is Ca^2+^ influx from the external medium, which is a trigger and essential for wound repair [21,22].

The Ca^2+^ influx leads to rapid membrane resealing. The ‘membrane patch hypothesis’ is proposed to plug the wound pore, wherein cytosolic membrane vesicles accumulate at the wound site and fuse with each other to form an impermanent ‘patch’ to plug the wound pore as an emergency reaction [22,23,24]. A recent research in *Xenopus* oocytes also supported this model by direct observation of the fusion of vesicle–vesicle and vesicle–cell membranes [25]. The source of membrane vesicles for the ‘patch’—such as lysosome [26,27] and cortical granules [22] have been proposed—but remains unclear.

A variety of hypotheses for wound repair that do not involve patching have also been proposed [2,28]. For example, the ‘exocytosis and endocytosis hypothesis’ involves the direct removal of the wound pore via exocytosis and subsequent endocytosis [29]. However, no clear consensus on the mechanism driving the repair process has been arrived at.

Annexins, a Ca^2+^-dependent membrane scaffold protein family, which bind to negatively charged membrane phospholipids in a Ca^2+^-dependent manner, have been implicated in muscle cell membrane repair [17,30,31]. Annexins also accumulate at the wound sites in other cells, and dysfunction of annexin inhibits the resealing process [15,32,33]. Endosomal sorting complex required for transport (ESCRT) has also been proved to be an essential component for membrane wound repair [34,35,36].

Cortical actin cytoskeleton is also rearranged at the wound site during wound repair [9,37,38]. In fruit fly embryos and frog oocytes, a contractile actomyosin ring is formed and its constriction closes the wound pore [11,39]. However, only actin accumulates at the wound site in smaller cells such as fibroblasts, yeast, and *Dictyostelium* cells, and not myosin II [12,40,41]. The actin assembly seems to be essential because a deletion of actin filaments leads to failure in the closing of the wound pore [9,39,42], but there is no direct evidence of wound repair, such as ceasing influx of outside dye, as far as we know.

Here, we examined wound repair in *Dictyostelium* cells by using a laserporation method, which we recently invented. We found for the first time that calmodulin plays an essential role in wound repair. We also found that actin accumulation at the wound site was dependent on Ca^2+^ influx and calmodulin, and it was essential for the wound repair. The membrane accumulated at the wound site to plug the wound pore by two-steps, depending on Ca^2+^ influx and calmodulin. From several lines of evidence, the membrane plug was derived from de novo generated vesicles at the wound site. We proposed a molecular mechanism of wound repair from the initial trigger to final closure of the wound pore. 

## 2. Materials and Methods

### 2.1. Cell Culture

*Dictyostelium discoideum* (AX2) and all mutant cells were cultured at 22 °C in a plastic dish containing HL5 medium (1.3% bacteriological peptone, 0.75% yeast extract, 85.5 mM e-glucose, 3.5 mM Na_2_HPO_4_, 3.5 mM KH_2_PO_4_, pH 6.3) [43]. For the wound experiments, HL5 medium was replaced with BSS (10 mM NaCl, 10 mM KCl, 3 mM CaCl_2_, and 3 mM MES, pH 6.3) and the cells were incubated in the same solution for 5–6 h.

### 2.2. Plasmids and Transformation

GFP-lifeact, GFP-actin, GFP-alpha-tubulin, GFP-cAR1, and annexin C1-GFP expression constructs have been previously described [20,44,45,46,47]. Full length pefA-GFP and vps4-GFP expression constructs were generated by cloning BamHI digested and inserting the PCR-amplified products into the pA15GFP vector. The GFP-lmpA expression construct was generated by cloning BamHI and SacI digested and inserting the PCR-amplified product into the GFP/pDNeo vector. GFP-calmodulin expression construct was obtained from NBRP Nenkin. Golvesin-GFP and GFP-calreticulin expression constructs were obtained from DictyBase. These expression constructs were transformed in cells by electroporation or laserporation, as described previously [47,48]. The transformed cells were selected in HL5 medium containing 10 µg/mL G418 (Wako Pure Chemical Corporation, Osaka, Japan) and/or 10 µg/mL blasticidin S hydrochloride (Wako Pure Chemical Corporation) in plastic dishes.

To generate the knockout constructs of the tsg101 or vsp4, each gene fragment was subcloned outside of two loxP sites in pLPBLP. The left (nucleotides 1–207) and right arms (nucleotides 334–563) of the tsg101 gene or the left (nucleotides 1–136) and right arms (nucleotides 355–525) of the vsp4 gene were amplified from the cDNA by PCR using the following primer sets with restriction enzyme sites (underscore): 5′-ATGTCGACATGTATGGTCATCATGGATACCCAATGC-3′ (SalI) and 5′-ATATCGATTGGTATATTTTCATAAAACGGTGATAAATTTGGG-3′ (ClaI) (for the left arm of the tsg101 KO construct); and 5′-ATGGATCCGATCCAACACCAGAGATGAGGATTGTAAAAAATC-3′ (BamHI) and 5′-ATACTAGTTATGGTGGTGGTGGTTGTTGTTGTTG-3′ (SpeI) (for the right arm of the tsg101 KO construct) or 5′-ATGTCGACATGGGTGATGTTAATTTCTTACAAAAAGCAATTC-3′ (SalI) and 5′-ATATCGATATTTTAAAGCTGTTGTAAACCATTCTAAACTTTGG-3′ (ClaI) (for the left arm of the vsp4 KO construct); and 5′-ATGGATCCGATTCTTTATCCTCTTCAATTGTAACAACAAAAC-3′ (BamHI) and 5′-ATACTAGTCTTACACCATATAATAAAATACCTTTCCATGG-3′ (SpeI) (for the right arm of the vsp4 KO construct). The amplified left arm fragments were subcloned between SalI and ClaI sites lying upstream of N-terminal loxP site in pLPBLP. The right arm fragments were subsequently subcloned between BamH1 and SpeI sites lying downstream of C-terminal loxP site. After transformation with the knockout constructs, the cells were selected in HL5 medium containing 10 µg/mL of blasticidin S hydrochloride.

### 2.3. Chamber Preparation

The surface of the coverslip of a glass-bottom chamber was coated with carbon by vapor deposition as previously described [20]. The thickness of the coat layer was approximately 20 nm. To make the surface hydrophilic, the surface of the coated coverslip was activated with plasma treatment. The chamber was sterilized with 70% ethanol and dried if necessary. The cells were settled on the surface of the coated coverslip, and they were mildly compressed with agarose block (1.5%, dissolved in BSS, 1 mm thick) to observe the ventral cell surface.

### 2.4. Wounding and Microscopy

The cells expressing GFP-proteins were observed under a total internal refection fluorescence microscope (TIRF, based on IX71 microscope, Olympus) as previously described [49]. Cells were wounded with a nanosecond–pulsed laser (FDSS532-Q, CryLas) and the wound size was set at 0.5 µm in diameter as previously described [20]. Time-lapse fluorescence images were acquired with 40–100 msec exposure times at 130–500 msec intervals using a cooled CCD camera (Orca ER, Hamamatsu Photonics). The time courses of fluorescence intensities were examined using Image J (http://rsbweb.nih.gov/ij). The fluorescence intensities were normalized by setting the value before wounding to 1 after subtracting the background.

### 2.5. Measurement of the Influx of Propidium Iodide, FM1-43, and Ca^2+^

To observe the influx of propidium iodide (PI) or FM1-43, 0.15 mg/mL of PI (Sigma-Aldrich, Tokyo, Japan) or 20 µM FM1-43 (Thermo Fischer Scientific, MA, USA) was included in the external medium. To monitor the dynamics of cytosolic Ca^2+^, cells expressing Dd–GCaMP6s were wounded using laserporation [20,50]. To observe recycling endosomes, after cells were stained with FM1-43 in BSS for 30 min, the dye bound to the cell membrane was removed by media exchange. These three probes were illuminated by an argon laser (488 nm) and monitored under TIRF microscopy. 

### 2.6. Ca–EGTA Buffer

To formulate medium containing the indicated free Ca^2+^, a Ca-EGTA buffer (10 mM KCl, 10 mM NaCl, 3 mM MES, 10 mM EGTA, and an appropriate concentration of CaCl_2_, pH 6.3) was used. The concentration of CaCl_2_ in the Ca-EGTA buffer was calculated by Ca–EGTA Calculator v1.3 (https://somapp.ucdmc.ucdavis.edu/pharmacology/bers/maxchelator/CaEGTA–TS.htm). The agarose block for the agar-overlay was made by dissolving 1.5% agarose in Ca-EGTA buffer or BSS as previously described [51].

### 2.7. Inhibitors

Latrunculin A (Funakoshi Co. Ltd., Tokyo, Japan) was dissolved in dimethyl sulfoxide (DMSO) to make a stock solution of 1 mM. The cells were incubated with a final concentration of 1 μM in BSS for 10 min before wound experiments. W7 hydrochloride (Funakoshi Co. Ltd.) was dissolved in DMSO to make a stock solution of 10 mM. The cells were incubated with a final concentration of 20 μM in BSS for 30 minutes before wound experiments. Thiabendazole (TB, Tokyo Chemical Industry Co. Ltd., Tokyo, Japan) was dissolved in DMSO to make a stock solution of 100 mM. To completely depolymerize the microtubules, the cells were incubated with a final concentration of 100 μM in BSS on ice for 30 min before wound experiments [44].

### 2.8. Statistical Analysis

Statistical analysis was performed using GraphPad Prism 7 (GraphPad software Inc., San Diego, CA, USA). Data are presented as the mean ± SD and analyzed using two-tailed Student’s *t*-test for comparison between two groups, or by one-way ANOVA with Tukey’s multiple comparisons test.

## 3. Results

### 3.1. Influx of Ca^2+^ through the Wound Pore is Essential for Wound Repair

We have reported an improved method to make wounds in the cell membrane by laserporation [20,47]. After the cells were placed on a coverslip coated with carbon by vapor deposition, a laser beam was focused on a small local spot beneath a single cell under a total internal reflection fluorescence (TIRF) microscope. The energy of the laser beam absorbed by the carbon made a small pore in the cell membrane that was attached to the carbon coat (Figure 1A).

Previously, we examined the opening of a pore in the cell membrane by laserporation in the presence of propidium iodide (PI), which emits fluorescence when it binds to RNA and DNA, in the external medium (BSS), a physiological saline that contains 3 mM Ca^2+^. Figure 1B shows a typical fluorescence microscopy image of PI influx through the wound pore. The fluorescence began to increase at the wound site and spread over the cytoplasm, which confirmed our previous observation [20]. Here, we also used FM1-43, which is a cell–impermeable fluorescent lipid analog that emits fluorescence when inserted into the membrane. Figure 1C shows a typical fluorescence microscopy of FM1-43 influx through the wound pore. The fluorescence also began to increase at the wound site and spread over the cytoplasm.

We previously showed that Ca^2+^ entered the cytoplasm through the wound pore from the external medium. To visualize the cytosolic Ca^2+^, the cells expressing Dd–GCaMP6s—a Ca^2+^ sensor consisting of GFP, calmodulin, and a peptide sequence of myosin light chain kinase (M13)—were wounded. The fluorescence increased from the wound site, spread over the cytoplasm, and finally decreased to a resting level (Figure 1D), which confirmed our previous observation [20]. Thus, the laserporation reliably creates a wound pore in the cell membrane.

To examine whether the Ca^2+^ influx is required for wound repair, we examined the effect of ethylene glycol-bis(β-aminoethyl ether)–*N*,*N*,*N*′,*N*′-tetraacetic acid (EGTA), a chelating agent of Ca^2+^, in the external medium. Figure 1E,F show typical fluorescence images of PI and FM1-43 influx, respectively. Figure 1H,I show the time course of the fluorescence intensities in the cells, indicating that in the presence of EGTA, PI, and FM influx did not cease after wounding whereas it ceased within a short time in BSS, suggesting that the wound pores do not close in the absence of Ca^2+^. In addition, the wounded cells showed no increase in fluorescence for Dd–GCaMP6s in the presence of EGTA (Figure 1G ,J). Therefore, we concluded that Ca^2+^ influx is essential for wound repair. Figure 1K,L show a dependency of PI influx on the external free Ca^2+^ concentration, suggesting that free Ca^2+^ concentration higher than 10^−4^ M in the external medium was required for the closure of wound pores.

### 3.2. Calmodulin, ESCRT, and Annexin Quickly Accumulate at the Wound Site

As the downstream of Ca^2+^ influx, annexin (a Ca^2+^-dependent membrane scaffolding protein) and ESCRT (endosomal sorting complexes required for transport) complexes have been identified in the wound repair of animal cells [29,52]. We previously showed that annexin C1-GFP accumulates at the wound site immediately after wounding, depending on the Ca^2+^ influx, but the annexin C1 knockout cells showed only a subtle defect of wound repair [20]. Incidentally, *Dictyostelium* cells have two annexin genes: annexin C1 and C2. Only annexin C1 accumulates at the wound site [20]. Here, we discovered that GFP-calmodulin, a Ca^2+^-binding messenger protein, and pefA-GFP and vps4-GFP, components of ESCRT, accumulated at the wound site immediately after wounding (Figure 2A).

Figure 2B shows time courses of the fluorescence intensity of GFP-calmodulin at the wound site in the presence (BSS) and absence of Ca^2+^ (EGTA). GFP-calmodulin did not accumulate in the absence of Ca^2+^, suggesting that its accumulation is dependent on Ca^2+^ influx. In addition, in the presence of W7, an inhibitor of calmodulin, GFP-calmodulin did not accumulate (Figure 2C). In the presence of W7, PI and FM1-43 influx did not cease after wounding, contrary to the control (Figure 2D,E). Therefore, calmodulin is essential for wound repair.

PefA-GFP and vps4-GFP did not accumulate in the presence of EGTA (Figure 2F,G), suggesting that their accumulation was dependent on Ca^2+^ influx. Vps4-GFP did not accumulate in the tsg101 (another component of ESCRT) null cells (Figure 2H). However, PI influx normally ceased after wounding in the tsg101 null (Figure 2I) and vps4 null cells (Figure 2J) as well as wild type cells. In addition, W7 did not inhibit GFP-pefA accumulation (Figure 2K). Therefore, ESCRT accumulates at the wound site dependent on Ca^2+^ influx and independent of calmodulin, and the ESCRT accumulation is not essential for wound repair.

Next, we examined the effect of W7 on annexin C1-GFP accumulation. W7 significantly inhibited GFP-annexin C1 accumulation (Figure 2L). In the knockout mutant of annexin C1, PI influx showed irregular responses but finally ceased (Figure 2M). Therefore, annexin C1 is also not essential for wound repair.

Together, although ESCRT, annexin C1, and calmodulin accumulate at the wound site depending on Ca^2+^ influx, among them, only calmodulin is essential for the wound repair in *Dictyostelium* cells.

### 3.3. Actin Accumulation at Wound Site is Essential for Wound Repair

We previously showed that actin accumulates at the wound site in *Dictyostelium* cells [41,47]. Figure 3A shows typical fluorescence images under TIRF microscopy when a cell expressing GFP-lifeact, a marker of actin filaments, was wounded. The actin filaments transiently accumulated at the wound site (arrows in Figure 3A). Since lifeact is an actin binding domain of yeast ABP140p, GFP-lifeact may not represent the exact timing of actin accumulation. Therefore, we also observed cells expressing GFP-actin (Figure 3B). Figure 3C,D show the time course of the relative fluorescence intensities of GFP-actin and GFP-lifeact at the wound site, respectively, indicating that they accumulated at the wound site with a similar time course (initiation time, peak time, and termination time, Figure 3E). When cells expressing GFP-lifeact were wounded in the presence of latrunculin A, depolymerizer of actin filaments, GFP-lifeact did not accumulate at the wound site in the presence of latrunculin A (Figure 3F). Therefore, actin accumulates in a filamentous form at the wound site.

Next, we examined whether Ca^2+^ influx is an upstream signal for the actin accumulation. Cells expressing GFP-lifeact were wounded in the presence of various external free Ca^2+^ concentration (Figure 3G), indicating that a free Ca^2+^ concentration higher than 10^−4^ M was required for actin accumulation, which is consistent with the free Ca^2+^ concentration necessary for the wound repair as shown in Figure 1L.

Next, we examined the role of actin accumulation on wound repair. After wounding, PI and FM1-43 influx did not cease in the presence of latrunculin A (Figure 3H and I). Therefore, we concluded that actin accumulation at the wound site is essential for wound repair.

In the presence of W7, actin accumulated but its amount was significantly reduced (Figure 3J). In addition, in the presence of latrunculin A, GFP-calmodulin normally accumulated (Figure 3K). Therefore, Ca^2+^ and calmodulin are a major upstream signal for the actin accumulation.

### 3.4. Membrane Plug is Formed at the Wound Site

To visualize the wound pore in the cell membrane, cells expressing GFP-cAR1 (cAMP receptor) as a marker of membrane protein, were wounded. Immediately after wounding, a black spot appeared at the position where laser was applied (Figure 4A). If the coverslip was not coated with carbon, such black spots did not appear, indicating that the black spot was not due to photobleaching by the laser beam but represented the wound pore. Figure 4B shows the time course of the relative size of the wound pore upon wounding. The wound pore transiently expanded slightly, then shrank, and finally closed within approximately 8 sec (8.14 ± 0.69 sec, *n* = 25). In the absence of Ca^2+^, the wound pore did not close and its size gradually increased (Figure 4B). In the presence of either latrunculin A or W7, the wound pore also did not close and its size gradually increased (Figure 4C,D). Therefore, we confirmed that actin filaments, Ca^2+^ influx, and calmodulin are essential for wound pore closure.

As shown in Figure 1C, FM1-43 spread over the cytoplasm after entering through the wound pore but some fluorescence remained at the wound site as a bright spot, suggesting that some membrane accumulates at the wound site, which may be the membrane to plug the wound pore. Interestingly, such membrane accumulation was not found in the presence of EGTA (Figure 1F), suggesting that the membrane plug is formed depending on the influx of Ca^2+^. The membrane plug may be derived from the accumulation and fusion of some cytosolic vesicles such as Golgi-derived vesicles, lysosome, endoplasmic reticulum, and recycling endosomes [2,53].

To identify the source of these vesicles, cells expressing golvesin-GFP, GFP-lmpA, GFP-calreticulin, and FM1-43 dye were used, respectively. Golvesin is a protein associated with the membranes of the Golgi apparatus and post-Golgi vesicles [54]. LmpA is a lysosomal-associated membrane glycoprotein [55,56]. Calreticulin is a Ca^2+^–binding protein present in the endoplasmic reticulum [57]. To visualize the recycling endosomes, 30 min after the cells were incubated with FM1-43, they were washed out by media exchange. Since the dye that stained the cell membrane was removed by washing, only stained internalized endosomes remained in the cells. Unexpectedly, none of them accumulated at the wound site, although golvesin-GFP transiently disappeared from the wound site upon wounding (Appendix A).

In the PI influx experiment (Figure 1B), PI fluorescence also remained at the wound site, suggesting that the cytoplasm including PI dye was entrapped in newly enclosed vesicles. Therefore, (1) broken cell membranes are enclosed to form vesicles including PI dye at the wound site, (2) vesicles are de novo generated at the wound site, or (3) unknown vesicles are generated at a distance and carried to the wound site.

### 3.5. Membrane Vesicles for the Plug are De Novo Generated at the Wound Site

Figure 4E shows typical fluorescence images at the wound site in FM1-43 influx experiments in the presence of BSS, EGTA, latrunculin A, and W7, respectively. Figure 4F shows time course of the fluorescence intensity of FM1-43 only at the wound site in the presence and absence of Ca^2+^. As mentioned above, the membrane accumulation (therefore, membrane plug formation) was dependent on Ca^2+^ influx.

In the presence of latrunculin A, the fluorescence intensity of FM1-43 at the wound site gradually increased compared with the control (Figure 4G). In the presence of W7, the fluorescence intensity of FM1-43 at the wound site also gradually increased (Figure 4H). Both the results suggest that the membrane plug is not derived from the broken cell membrane because the amount of the broken cell membrane should be limited and, therefore, it is unlikely that an increasing amount of membrane accumulates at the wound site.

If the source of the membrane plug is unknown vesicles generated at a distance and carried to the wound site, actin filaments or microtubules must contribute to their translocation. Since the amount of membrane gradually increased at the wound site in the presence of latrunculin A (Figure 4G), it is unlikely that actin filaments contribute to carrying the vesicles. Actin filaments actually restrained the level of membrane accumulation.

Microtubules accumulate at the wound site in *Xenopus* oocytes [58], and elongate toward the wound site in epithelial kidney cells [59]. When *Dictyostelium* cells expressing GFP-tubulin were wounded, microtubules neither accumulated nor elongated toward the wound site upon wounding (Figure 4I). Figure 4J shows typical fluorescence images at the wound site using FM1-43, cells expressing annexin C1-GFP, and PI in the presence of thiabendazole (TB), a depolymerizer of microtubules. The membrane normally accumulated at the wound site in FM1-43 experiments (Figure 4K), and experiments using annexin C1-GFP and PI also showed similar curves to those in the absence of TB (Figure 4L,M). Therefore, the vesicles are not carried along microtubules and microtubules are not necessary for wound repair.

Thus, it is plausible that the membrane at the wound site is not carried from other places, but vesicles are de novo generated near the wound site.

## 4. Discussion

In the present study, we used our recently invented laserporation method to make a wound in *Dictyostelium* cells, and found that annexin C1, calmodulin, and ESCRT components quickly accumulated at the wound site. Annexin and ESCRT have been reported to contribute to wound repair in several model organisms [15,29,36,52,60,61,62]. However, in *Dictyostelium* cells, ESCRT did not contribute to the wound repair while annexins partially contributed. In the present study, calmodulin is considered as a new member for the single cellular wound repair signaling, although it has been reported to play a role in multicellular wound repair in plant cells [63].

The calmodulin inhibitor inhibited both actin and annexin C1 accumulation. However, since this inhibition is partial, another unknown, calmodulin-independent pathway may exist. As calmodulin is not so lipophilic as to directly attach to the cell membrane, its target protein must bind to the wounded cell membrane, depending on Ca^2+^/calmodulin. The target protein remains to be found in future.

We also elucidated the dynamics of membrane plug formation from the experiments using FM1-43 and PI dyes. The staining with FM1-43 remained as a fluorescent dot at the wound site even after its spreading over the cytoplasm, indicating that some membrane accumulated at the wound site, representing a membrane plug. We searched for the source of this membrane using markers of Golgi-derived vesicles, lysosome, endoplasmic reticulum, and recycling endosomes, but none of them accumulated at the wound site. In addition, PI fluorescence also remained at the wound site (Figure 1B), suggesting that the cytoplasm including PI dye was entrapped in newly enclosed vesicles. Therefore, it is unlikely that preexisting vesicles are the source of the membrane plug. Then, we speculated three possible sources for the membrane plug: broken cell membranes, de novo generated vesicles at the wound site, and unknown vesicles that are carried to the wound site.

It is unlikely that the membrane plug is derived from the broken cell membrane because the amount of the broken cell membrane is limited. Active transport of intracellular vesicles to the wound site has been reported to supply the membrane to plug the wound pore in animal cells [22,64,65]. However, given the high speed of the membrane plug formation, it is unlikely that vesicles are carried from other places in *Dictyostelium* cells, which is, in addition, supported by the observations that microtubules and actin inhibitors did not hinder the membrane accumulation.

Therefore, the vesicles for the membrane plug are de novo generated at the wound site. At present, the mechanism for the generation of vesicles de novo is unclear. The cell membrane components may flow toward the wound site to supply the membrane to generate the vesicles. Although we did not observe flow of GFP-cAR1 toward the wound site, which could be observed as single molecules under TIRF microscopy [66], we cannot rule out the possibility that lipids and other membrane proteins than cAR1 can laterally move in the cell membrane toward the wound edge and supply the membrane to generate vesicles. Otherwise, vesicles are completely de novo generated near the wound site by an unknown mechanism. Incidentally, autophagic vacuoles are also de novo generated, although the mechanism is elusive [67]. We need to directly observe the generation of vesicles and their plug formation by electron microscopy in future.

Several models for wound repair have been proposed, including membrane patch, exocytosis, endocytosis, and plugging wound pores with vesicle aggregates [22,23,25,26,27,29]. Most models rely on the preexisting compartments such as lysosomes and cortical granules, which are not consistent with the present results. The present observation is basically consistent with the membrane patch hypothesis, except that the vesicles for membrane plug are de novo generated.

A recent excellent research in *Xenopus* oocytes directly observed the fusion of vesicle–vesicle and vesicle–cell membrane upon wounding, whereas the vesicles were cortical granules specific to oocytes [25]. In their observations, the fluorescent dextran flowing from the wound pore was confined at the wound site, which is consistent with our observations (PI entrapping). Although these authors discussed that the dextran diffusion was spatially limited by the tightly packed vesicles, we prefer our de novo generation model because PI molecules are highly diffusible to spread over the cytoplasm.

From the kinetics of experiments using FM 1–43 in the presence of latrunculin A (Figure 4G), the membrane accumulated at the wound site by two steps: the initial rapid accumulation upon wounding and thereafter gradually increasing accumulation. The second accumulation was also observed in a very subtle amount in the control (Figure 4F). Therefore, we propose that membrane plug formation occurs by two steps, urgent membrane plug formation and following plug completion. In the presence of W7 or latrunculin A, PI influx and FM1-43 influx did not cease (Figure 2D,E, Figure 3H,I), suggesting that the urgent membrane plug is incomplete, and additional mechanisms such as actin accumulation and calmodulin-mediated unknown mechanism are required to completely close the wound pore.

Figure 5A shows a summary of the duration time of each probe. The wound pore is estimated to close within approximately 8 sec from direct observations of the wound pore using cells expressing GFP-cAR1, and for approximately 2–3 sec from indirect observations of FM1-43 influx and PI influx. Since the entry of GFP-cAR1 molecules into the membrane plug by diffusion may take a longer time due to the hindrance by the membrane plug components, it is plausible that the closing time by the urgent membrane plug is estimated at around 2–3 sec. Following this urgent membrane plug, actin began to accumulate at around 2.5 sec, which was much later than the initiation times of the urgent membrane plug formation and calmodulin accumulation. The membrane plug will become complete at the termination time of actin accumulation (~20 sec).

One of the roles of actin accumulation may be to prevent the pore from expanding furthermore or to actively contribute to the constriction of the wound pore as discussed previously [9,42]. In the present study, in the presence of latrunculin A, the size of the membrane pore gradually increased, and an increasing amount of membrane accumulated, suggesting an additional role for actin accumulation to prevent further membrane accumulation, although the detailed mechanism is elusive.

In *Xenopus* oocytes, the cortical actin filaments accumulate toward the wound site by their flow [68]. In spite of the ability to observe individual actin filaments under TIRF microscopy, we did not observe such flow around the wound site. Presumably, actin de novo polymerizes at the wound site. The mechanism by which actin accumulates at the wound site is to be clarified in future.

Annexins have been reported to promote wound repair of animal cells by fusing intracellular vesicle to the cell membrane [33,69]. In addition, annexins appear to restrict wound expansion and contract the wound edge by their assembly to 2D-ordered arrays [15,16]. At present, we do not know the exact role of *Dictyostelium* annexin C1 for wound repair, but the observations that W7 inhibited its accumulation at the wound site and PI showed irregular influx in annexin C1 null cells indicate that annexin C1 may partially contribute to both urgent and second steps of the plug formation.

Figure 5B and C show a summary of wound repair in *Dictyostelium* cells. Upon wounding, the influx of Ca^2+^ through the wound pore is a signal to form the urgent membrane plug at the wound site, where membrane vesicles are de novo generated, followed by mutual fusion of vesicle–vesicle and vesicle–cell membranes. The Ca^2+^ influx also causes calmodulin and annexin C1 to accumulate at the wound site. In addition, actin accumulates by de novo polymerization to complete the membrane plug. The actin accumulation depends on Ca^2+^ influx and calmodulin. From Figure 5A, the initiation of the actin disassembly may be mediated by the disengagement of calmodulin from the wound site. The signals and effectors for the actin accumulation remain to be clarified going forward.

In the present study, the membrane plug remained without disappearing during observations. We previously observed that the wound region is finally shed on the surface of the coverslip as the cell migrated away [50]. Therefore, the remaining membrane plug appears to not return to the unwounded state in *Dictyostelium* cells although it has been thought to return to the unwound state or to be endocytosed in animal cells [2,70,71].

## Figures and Tables

**Figure 1 cells-09-01058-f001:**
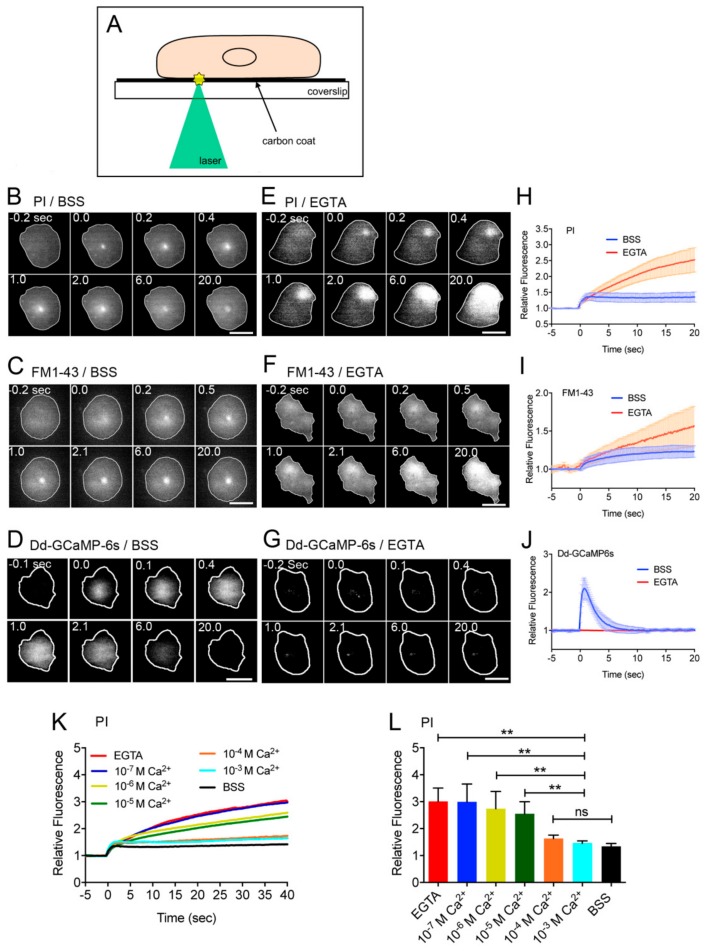
Ca^2+^ influx from the wound pore is essential for the wound repair. (**A**) To make a wound in the cell membrane, after cells were placed on a carbon-coated coverslip, a laser beam was focused on a small local spot beneath a single cell under a TIRF microscope. The wound size was set at 0.5 µm in diameter. (**B**,**E**) Typical sequences of fluorescence images of propidium iodide (PI) influx after laserporation in the presence (**B**) and absence (**E**) of Ca^2+^. The cells (outlined with white line) were wounded at 0 sec. (**C**,**F**) Typical sequences of fluorescence images of FM1-43 influx after laserporation in the presence and absence of Ca^2+^. (**D**,**G**) Typical sequences of fluorescence images of cells expressing Dd–GCaMP6s after laserporation in the presence and absence of Ca^2+^. (**H**) Time course of PI influx in the presence and absence of Ca^2+^. (**I**) Time course of FM1-43 influx after laserporation in the presence and absence of Ca^2+^. (**J**) Time course of the fluorescence intensity of cells expressing Dd–GCaMP6s after laserporation in the presence and absence of Ca^2+^. (**K**) Time courses of PI influx in the various free Ca^2+^concentrations in the external medium. (**L**) A dependency of the PI influx on the free Ca^2+^ concentration in the external medium. The relative fluorescence intensities 40 sec after wounding were plotted versus each Ca^2+^ concentration. Data are presented as mean ± SD (n = 25, each). ** *p* ≤ 0.0001; ns, not significant, *p* > 0.05. Bars, 10 µm.

**Figure 2 cells-09-01058-f002:**
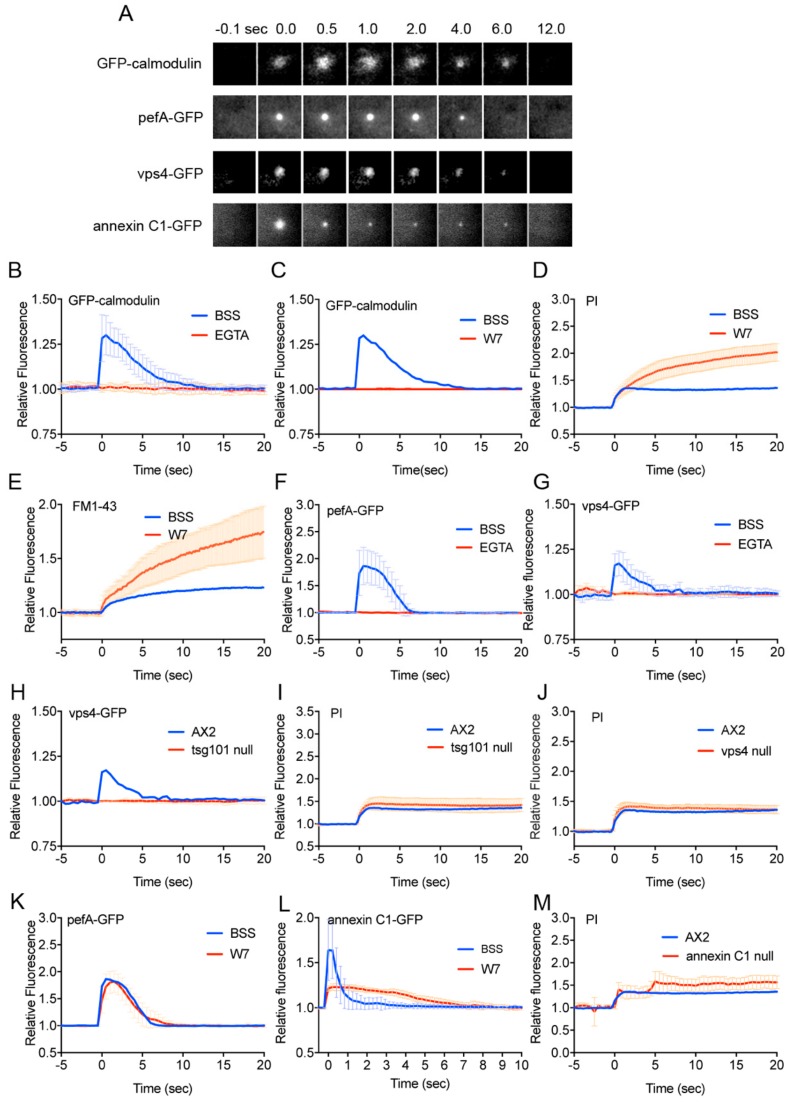
Calmodulin, ESCRT, and annexin quickly accumulate at the wound site. (**A**) Typical sequences of fluorescence images at the wound sites in cells expressing GFP-calmodulin, pefA-GFP, vsp4-GFP, and annexin C1-GFP. (**B**) The time course of fluorescence intensity of GFP-calmodulin at the wound site in the presence and absence of Ca^2+^. (**C**) The time course of GFP-calmodulin in the presence and absence of W7. (**D**) The time course of PI influx in the presence and absence of W7. (**E**) The time course of FM1-43 influx in the presence and absence of W7, an inhibitor of calmodulin. (**F**) The time course of fluorescence intensity of pefA-GFP in the presence and absence of Ca^2+^. (**G**) The time course of fluorescence intensity of vps4-GFP in the presence and absence of Ca^2+^. (**H**) The time course of fluorescence intensity of vps4-GFP in tsg101 null and wild type cells. (**I**) The time course of PI influx in tsg101 null and wild type cells. (**J**) The time course of PI influx in vps4 null and wild type cells. (**K**) The time course of fluorescence intensity of pefA-GFP in the presence and absence of W7. (**L**) The time course of fluorescence intensity of annexin C1-GFP in the presence and absence of W7. (**M**) The time course of PI influx in annexin C1 null and wild type cells. All graphs are shown with average ± SD (n = 25, each).

**Figure 3 cells-09-01058-f003:**
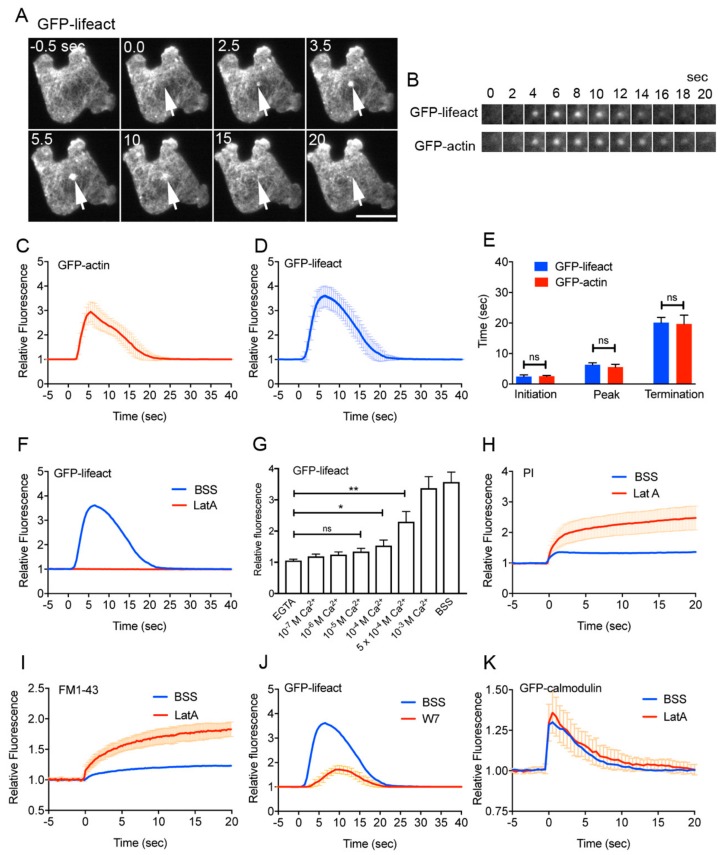
Actin accumulation is essential for wound repair. (**A**) A typical sequence of fluorescence images of a cell expressing GFP-lifeact, as a marker of actin filaments. Arrows indicate the position of wound site. Bar, 10 µm. (**B**) Typical sequences of fluorescence images at the wound sites of cells expressing GFP-lifeact and GFP-actin, respectively. (**C**) Time course of relative fluorescence intensity of GFP-actin at wound site. (**D**) Time course of relative fluorescence intensity of GFP-lifeact at wound site. (**E**) Comparison of initiation time, peak time, and termination time between GFP-lifeact and GFP-actin. (**F**) Time courses of relative fluorescence intensity of GFP-lifeact in the presence and absence of latrunculin A. (**G**) A dependency of the accumulation of GFP-lifeact at the wound site on free Ca^2+^ concentration in the external medium. The peak intensities of GFP-lifeact were plotted versus each free Ca^2+^ concentration. Data are presented as mean ± SD (n = 25, each). * *p* ≤ 0.001; ** *p* ≤ 0.0001; ns, not significant, *p* > 0.05. (**H**) Time courses of PI influx in the presence and absence of latrunculin A, a depolymerizer of actin filaments. (**I**) Time courses of FM1-43 influx in the presence and absence of latrunculin A. (**J**) Time courses of relative fluorescence intensity of GFP-lifeact in the presence and absence of W7. (**K**) Time courses of relative fluorescence intensity of GFP-calmodulin in the presence and absence of latrunculin A. All graphs are shown with average ± SD (n = 25, each).

**Figure 4 cells-09-01058-f004:**
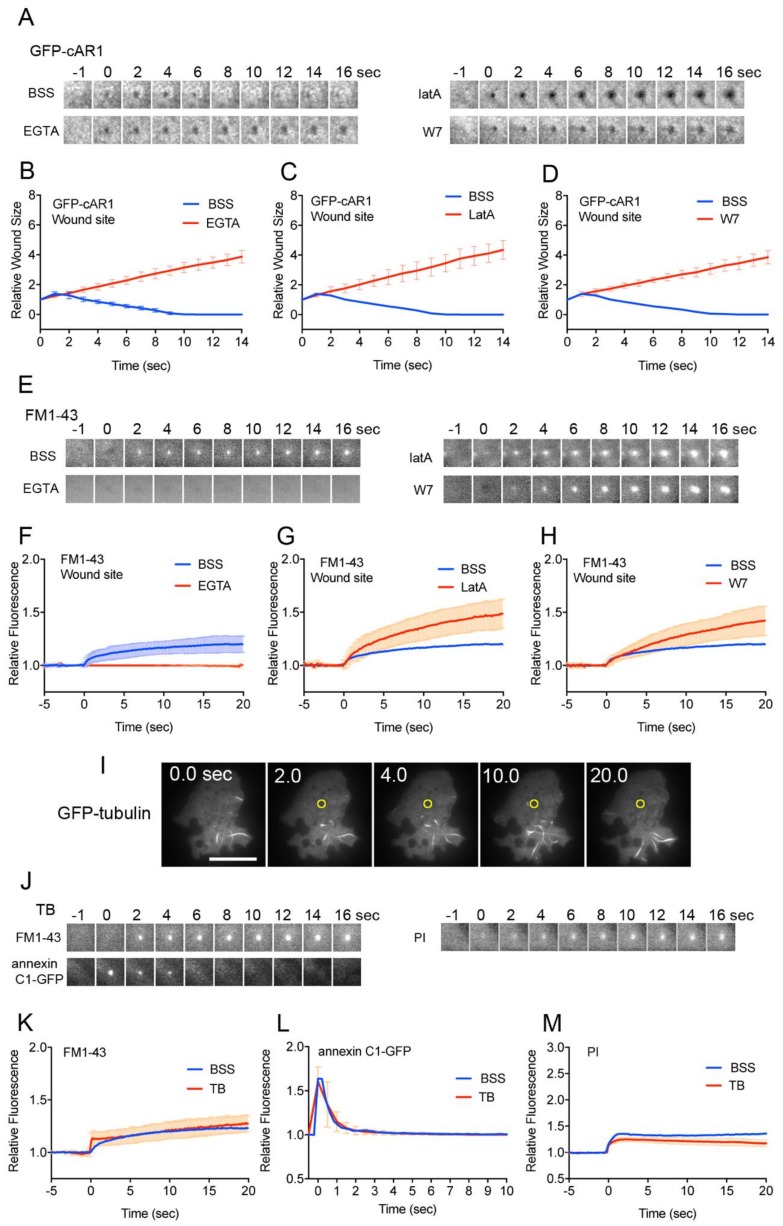
The membrane vesicles for the plug are de novo generated at the wound site. (**A**) Typical sequences of fluorescence image at the wound sites of cells expressing GFP-cAR1 in the presence of BSS, EGTA, latrunculin A, and W7. The wound pore was visualized as a black spot. (**B**) Time courses of the size (diameter) of black spots in the presence and absence of Ca^2+^. (**C**) Time courses of the black spot size in the presence and absence of latrunculin A. (**D**) Time courses of the black spot size in the presence and absence of W7. (**E**) Typical fluorescence images at the wound site in FM1-43 influx experiments in the presence of BSS, EGTA, latrunculin A, and W7, respectively. (**F**) Time courses of the fluorescence intensity of FM1-43 at the wound site in the presence and absence of Ca^2+^. (**G**) Time courses of the fluorescence intensity of FM1-43 at the wound site in the presence and absence of latrunculin A. (**H**) Time courses of the fluorescence intensity of FM1-43 at the wound site in the presence and absence of W7. (**I**) Typical fluorescence images of wounded cells expressing GFP-tubulin. Yellow circles show the wound site. Note that only fragmented images of microtubules were observed under TIRF microscopy. (**J**) Typical sequences of fluorescence image at the wound sites in the presence of TB, a depolymerizer of microtubules; experiments using FM1-43, annexin C1-GFP, and PI, respectively. (**K**) Time courses of the fluorescence intensity of FM1-43 at the wound site in the presence and absence of TB. (**L**) Time courses of the fluorescence intensity of annexin C1-GFP at the wound site in the presence and absence of TB. (**M**) Time courses of the fluorescence intensity of PI at the wound site in the presence and absence of TB. All graphs are shown with average ± SD (n = 25, each).

**Figure 5 cells-09-01058-f005:**
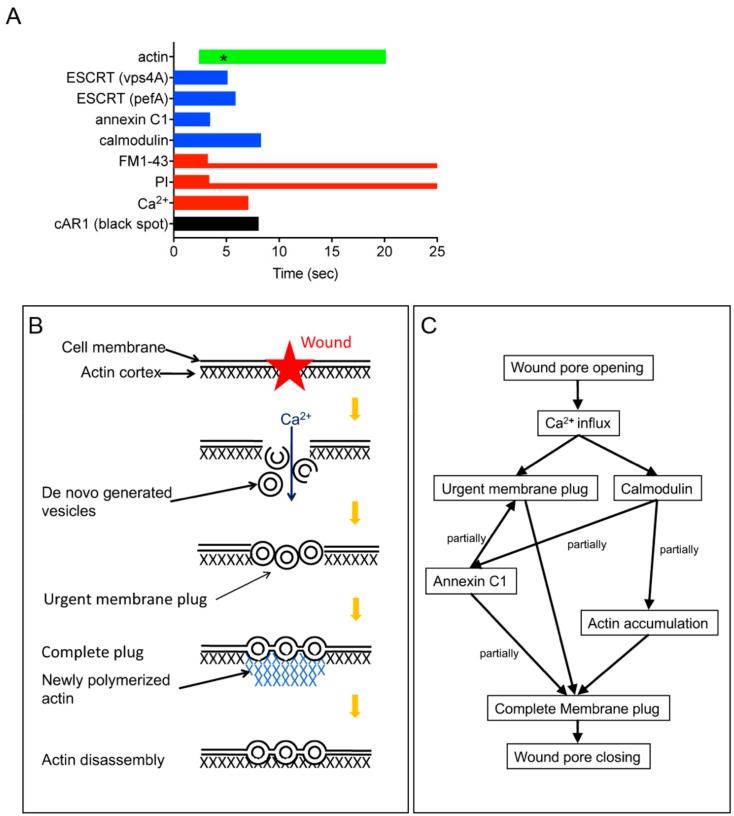
A summary of wound repair in *Dictyostelium* cells. (**A**) A summary of the duration time of each probe. Aster (black) in the actin bar (green) indicates the peak of actin accumulation. FM1-43 and PI accumulate at the wound site for 2–3 sec, but their fluorescence remained as a membrane plug during observation. (**B**) A schema for a wound repair mechanism. Upon wounding, Ca^2+^ enters through the wound pore and trigger the de novo generation of vesicles and mutual fusion of vesicle–vesicle and vesicle–cell membrane to make an urgent membrane plug. Actin accumulates to complete the plug, depending on Ca^2+^ and calmodulin. (**C**) A signal pathway for the wound repair, estimated from the present results.

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
