# Peer review of "Ca^2+^–Calmodulin Dependent Wound Repair in *Dictyostelium* Cell Membrane"

_cells, 2020, doi:10.3390/cells9041058_

Round 1

Reviewer 1 Report

In the manuscript titled “Ca2+-Calmodulin Dependent Wound Repair in 2 Dictyostelium Cell Membrane”, the authors aimed to study wound repair in Dictyostelium cells and the role of cytosolic Ca2+, annexin, 20 calmodulin, and endosomal sorting complexes required for transport (ESCRT) components.  They then proposed a molecular mechanism of wound repair.

The manuscript is written and presented with very high quality, and the science is presented very clearly.  The approach was well thought through, and systematically performed and presented.

The reason behind the study was clearly defined and presented, and the subsequent conclusions and proposed mechanism appears scientifically sound.

Very minor word choice suggestions were made on the manuscript.

Author Response

First of all, we would like to greatly appreciate your valuable reviewing and suggestions.

Line 25 and other parts: ‘de novo’ is changed to italic.

Line 33: ‘an’ is delated.

Line 59: efficient --> essential

Line 401: conducts --> occurs

Reviewer 2 Report

The authors studied wound repair in Dictyostelium cells using a novel laserporation. They carried out fine experiments with the use of fluorescent dyes and GFP-labelled proteins to monitor kinetics of the processes of wound repair. They found that annexin, calmodulin, and endosomal sorting complexes accumulate at the wound site upon wounding. They revealed that membrane accumulate at the wound site to plug the wound pore by two-steps, depending on Ca2+ influx and calmodulin. Actin filaments also accumulate at the wound site, depending on Ca2+ influx and calmodulin. The authors suggested a reasonable molecular mechanism of wound repair. The work is performed at a high experimental level.

Author Response

We would like to greatly appreciate your valuable reviewing.

Reviewer 3 Report

Revision report

In the manuscript entitled ‘Ca2+-Calmodulin Dependent Wound Repair in Dictyostelium Cell Membrane’, the authors Talukder et al. further investigate the mechanism of cell membrane wound repair in Dictyostelium they previously analyzed with the same technique in a previous work on Sci. Rep (2018).

Although the technique they have developed was firstly described in 2016 (Yumura S. A novel low-power laser-mediated transfer of foreign molecules into cells. Sci Rep. 2016 Feb 23;6:22055. doi: 10.1038/srep22055. PubMed PMID:26902313; PubMed Central PMCID: PMC4763237), the authors still claim its novelty. Actually, though not a pure novelty, the technique is surely at the highest standard. Although some of the aspects of the process they describe have already been described earlier (requirement for Ca2+ influx, involvement of annexin C1 and of ESCRT complex components, cortical actin skeleton rearrangement), the manuscript describes novel features of the wound healing process, and this process is actually far from clarified yet. For example, the authors demonstrate for the first time the requirement for calmodulin for wound repair, and its involvement in actin, annexin C1, and ESCRT complex recruitment to the wound site, but the dispensability of these last two for the repairing process. Moreover, de novo synthesis of vesicles is proposed at the wound site as the mechanism for the repair through a membrane plug, since no presence of Golgi or lysosome markers is detected in the membrane plug and no microtubules are assembled to drive transport of vesicles to the damage site. Thus, the manuscript is original and the results surely provide an advance in current knowledge on a scarcely known but ubiquitous biological process. The results appear significant, scientifically sound, and technically advanced, although their impact should be better focused. The manuscript is in my opinion to be considered for publication, as long as some improvements are made in its writing and some missing data are provided.

General remarks

Abstract section. Though it is worthy to describe the known characteristics and requirements of the process in the abstract section, the authors should rewrite the abstract in order to better underline here which is the novelty of the presented results, such as the role of calmodulin in wound repair and actin filament recruitment and the de novo synthesis of vesicles to repair the wound, while clearly describing the other features as background.

-Results section. Some results are presented as novelty but they were already described previously. The authors should carefully revise the results section and clearly state which results are confirmation of previous data (such as the inhibition of wound repair by EGTA in the external medium, already reported in the same authors paper on Sci Rep 2018) and which ones are genuine novelty. For instance, the point of the section 1.1 should be the assessment of the minimal concentration of external Ca2+ required for wound repair, not the mere requirement for Ca2+ presence.

-The laserporation method, though technically advanced, is not novel. It was described for the first time in 2016, and was already used in a previous work with remarkable results. The authors should mention that, but not claim the novelty of the method in this work.

-Discussion. The author should further discuss the involvement of ESCRT complex component. It is not clear if the loss of one of the components is enough to assess the dispensability of the full complex. It is not clearly discussed why it was impossible to knock out one of the components but it was possible to knock out other two. Moreover, annexin C1 lack also did cause only a minor defect in wound repair. The involvement of these unessential proteins in the process should be discussed. Finally, the frame of previous results with respect to these novel findings is not sufficiently treated and it is not clearly stated if the reported data are in agreement with previous findings from different cellular models or if they are a totally novel breakthrough in the field. This impairs to appreciate the real impact of the work and it should be clearly discussed in this section but also better evidenced in the abstract.

Major remarks

-The authors aim to assess which could be the origin of the vesicles that coalesce in the membrane plug. They perform tests in order to assess if they are originated from the Golgi of from lysosomes. Though it is understandable that the origin from the plasma membrane itself is unlikely, it was actually previously suggested that the origin of the membrane plug would be the fusion of vesicles derived from peripheral ER compartment (Mellgren RL. A plasma membrane wound proteome: reversible externalization of intracellular proteins following reparable mechanical damage. J Biol Chem. 2010; 285:36597–607. [PubMed: 20810652]). The authors should provide some data either to confirm or dispute this affirmation, for example using a ER resident protein fused to GFP as they have done for the Golgi marker.

Minor remarks

-Legend to Figure 2 should contain a description of the action of W7.

-Legend to Figure 3 should contain a description of GFP-lifeact and latrunculin A functions.

-Legend to Figure 4 should contain a description of the action of TB.

-Lines 362-363. The sentence need to be fixed

-Lines 381-382. The sentence need to be fixed

Author Response

First of all, we would like to greatly appreciate your valuable reviewing and suggestions.

C1: Abstract section. Though it is worthy to describe the known characteristics and requirements of the process in the abstract section,the authors should rewrite the abstract in order to better underline here which is the novelty of the presented results, such as the role of calmodulin in wound repair and actin filament recruitment and the de novo synthesis of vesicles to repair the wound, while clearly describing the other features as background.

A1: We tried to improve the summary according the reviewer’s suggestion.

Wound repair of cell membrane is a vital physiological phenomenon. We examined wound repair in Dictyostelium cells by using a laserporation, which we recently invented. We examined the influx of fluorescent dyes from the external medium and monitored the cytosolic Ca2+ after wounding. The influx of Ca2+ through the wound pore was essential for wound repair. Annexin and ESCRT components accumulated at the wound site upon wounding as previously described in animal cells, but these were not essential for wound repair. We newly found that calmodulin accumulated at the wound site upon wounding, which was essential for wound repair. The membrane accumulated at the wound site to plug the wound pore by two-steps, depending on Ca2+ influx and calmodulin. From several lines of evidence, the membrane plug was derived from de novo generated vesicles at the wound site. Actin filaments also accumulated at the wound site, depending on Ca2+ influx and calmodulin. Actin accumulation was essential for wound repair, but microtubules were not essential. A molecular mechanism of wound repair will be discussed.

C2: Results section. Some results are presented as novelty but they were already described previously. The authors should carefully revise the results section and clearly state which results are confirmation of previous data (such as the inhibition of wound repair by EGTA in the external medium, already reported in the same authors paper on Sci Rep 2018) and which ones are genuine novelty. For instance, the point of the section 1.1 should be the assessment of the minimal concentration of external Ca2+ required for wound repair,not the mere requirement for Ca2+ presence.

A2: Some data are presented to be just confirmed but essential for the presentation of following results. We would like to clearly state confirmation of our previous results for these data.

Figure 1 is not just the assessment of the minimal concentration of external Ca2+ required for wound repair. Influx of Ca2+ from the wound pore is essential for the wound repair.

C3: The laserporation method, though technically advanced, is not novel. It was described for the first time in 2016, and was already used in a previous work with remarkable results. The authors should mention that, but not claim the novelty of the method in this work.

A3: We would like to remove the modifying term ‘novel’ about the laserporation in the text.

C4: Discussion. The author should further discuss the involvement of ESCRT complex component. It is not clear if the loss of one of the components is enough to assess the dispensability of the full complex. It is not clearly discussed why it was impossible to knock out one of the components but it was possible to knock out other two.

A4: Very recently, we succeeded to make a knockout mutant of vps4 and found that PI influx of this mutant ceased normally. This observation also indicated that ESCRT is not essential. We would like to add this data in Figure 2.

C5: Moreover, annexin C1 lack also did cause only a minor defect in wound repair. The involvement of these unessential proteins in the process should be discussed.

A5: We would like to add a discussion about the rule of annexin C1 as follows:

Line 190: Annexins have been reported to promote wound repair of animal cells by fusing intracellular vesicle to the cell membrane{Lennon et al., 2003, #15809; McNeil et al., 2006, #54190}. In addition, annexins appear to restrict wound expansion and contract the wound edge by their assembly to 2D-ordered arrays {Bouter et al., 2011, #75249; Boye et al., 2017, #7334}. At present, we do not know the exact role of Dictyostelium annexin C1 for wound repair, but the observations that W7 inhibited its accumulation at the wound site and PI showed irregular influx in annexin C1 null cells indicate that annexin C1 may partially contribute to both urgent and second steps of the plug formation.

C6: Finally, the frame of previous results with respect to these novel findings is not sufficiently treated and it is not clearly stated if the reported data are in agreement with previous findings from different cellular models or if they are a totally novel breakthrough in the field. This impairs to appreciate the real impact of the work and it should be clearly discussed in this section but also better evidenced in the abstract.

A6: We would like to add an explanation that the reported data are not in agreement with previous findings from different cellular models, as follows.

Several models for wound repair have been proposed, including membrane patch, exocytosis, endocytosis, and plugging wound pores with vesicle aggregates [22,23,25,26,27,29]. Most of models rely on the preexisting compartments such as lysosomes and cortical granules, which is not consistent with the present results. The present observation is basically consistent with the membrane patch hypothesis, except that the vesicles for membrane plug are de novo generated.

A recent excellent research in Xenopus oocytes directly observed the fusion of vesicle-vesicle and vesicle-cell membrane upon wounding, whereas the vesicles were cortical granules specific to oocytes [25]. In their observations, the fluorescent dextran flowing from the wound pore was confined at the wound site, which is consistent with our observations (PI entrapping). Although these authors discussed that the dextran diffusion was spatially limited by the tightly packed vesicles, we prefer our de novo generation model because PI molecules are highly diffusible to spread over the cytoplasm.

C7: Major remarks: The authors aim to assess which could be the origin of the vesicles that coalesce in the membrane plug. They perform tests in order to assess if they are originated from the Golgi of from lysosomes. Though it is understandable that the origin from the plasma membrane itself is unlikely, it was actually previously suggested that the origin of the membrane plug would be the fusion of vesicles derived from peripheral ER compartment (Mellgren RL. A plasma membrane wound proteome: reversible externalization of intracellular proteins following reparable mechanical damage. J Biol Chem. 2010; 285:36597–607. [PubMed: 20810652]). The authors should provide some data either to confirm or dispute this affirmation, for example using a ER resident protein fused to GFP as they have done for the Golgi marker.

A7: We would like to cite the reference that the reviewer suggested and add new data using a marker of ER in the Fig.S1, indicating that ER is not the source of the membrane plug. We also added an explanation of the plasmid for this ER marker, and inserted related sentence in the text.

C8: Minor remarks: Legend to Figure 2 should contain a description of the action of W7.

A8: The description of the action of W7 was added.

C9: Legend to Figure 3 should contain a description of GFP-lifeact and latrunculin A functions.

A9: The descriptions of the GFP-lifeact and latrunculin A were added.

C10: Legend to Figure 4 should contain a description of the action of TB.

A10: The description of the action of TB was added.

C11: Lines 362-363. The sentence need to be fixed

A11: We would like to change the sentence as follows.

However, in Dictyostelium cells, while annexins partially contributed, but ESCRT did not contribute to the wound repair. --> However, in Dictyostelium cells, ESCRT did not contribute to the wound repair while annexins partially contributed.

C12: Lines 381-382. The sentence need to be fixed

A12: The mistake is fixed

It is unlikely that the membrane plug is not derived from the broken cell membrane --> It is unlikely that the membrane plug is not derived from the broken cell membrane

Round 2

Reviewer 3 Report

The present version of the manuscript by Talukder et al. has solved all of the issues raised during my first revision. The presentation of the data has been significantly improved, due to further analysis and discussion of some points. The manuscript surely contributes to knowledge in the field by adding new features to the still not fully understood process off membrane wounds repair.

I reccomand publication of the manuscript in the present form.